# Chimpanzee brain morphometry utilizing standardized MRI preprocessing and macroanatomical annotations

Sam Vickery[1,2]*, William D Hopkins[3], Chet C Sherwood[4], Steven J Schapiro[3,5], Robert D Latzman[6], Svenja Caspers[7,8,9], Christian Gaser[10,11], Simon B Eickhoff[1,2], Robert Dahnke[10,11,12]†*, Felix Hoffstaedter[1,2]†*

[1]Institute of Systems Neuroscience, Medical Faculty, Heinrich-Heine-University, Düsseldorf, Germany; [2]Institute of Neuroscience and Medicine (INM-7) Research Centre Jülich, Jülich, Germany; [3]Keeling Center for Comparative Medicine and Research, The University of Texas MD Anderson Cancer Center, Bastrop, United States; [4]Department of Anthropology and Center for the Advanced Study of Human Paleobiology, The George Washington University, Washington, United States; [5]Department of Experimental Medicine, University of Copenhagen, Copenhagen, Denmark; [6]Department of Psychology, Georgia State University, Atlanta, United States; [7]Institute of Neuroscience and Medicine (INM-1), Research Centre Jülich, Jülich, Germany; [8]Institute for Anatomy I, Medical Faculty, Heinrich-Heine-University, Düsseldorf, Germany; [9]JARA-BRAIN, Jülich-Aachen Research Alliance, Jülich, Germany; [10]Structural Brain Mapping Group, Department of Neurology, Jena University Hospital, Jena, Germany; [11]Structural Brain Mapping Group, Department of Psychiatry and Psychotherapy, Jena University Hospital, Jena, Germany; [12]Center of Functionally Integrative Neuroscience, Department of Clinical Medicine, Aarhus University, Aarhus, Denmark

*For correspondence:
s.vickery@fz-juelich.de (SV);
robert.dahnke@uni-jena.de (RD);
f.hoffstaedter@fz-juelich.de (FH)

†These authors contributed
equally to this work

Reviewing editor: Jonathan Erik
Peelle, Washington University in
St. Louis, United States

**Abstract** Chimpanzees are among the closest living relatives to humans and, as such, provide a crucial comparative model for investigating primate brain evolution. In recent years, human brain mapping has strongly benefited from enhanced computational models and image processing pipelines that could also improve data analyses in animals by using species-specific templates. In this study, we use structural MRI data from the National Chimpanzee Brain Resource (NCBR) to develop the chimpanzee brain reference template Juna.Chimp for spatial registration and the macro-anatomical brain parcellation Davi130 for standardized whole-brain analysis. Additionally, we introduce a ready-to-use image processing pipeline built upon the CAT12 toolbox in SPM12, implementing a standard human image preprocessing framework in chimpanzees. Applying this approach to data from 194 subjects, we find strong evidence for human-like age-related gray matter atrophy in multiple regions of the chimpanzee brain, as well as, a general rightward asymmetry in brain regions.

## Introduction

Chimpanzees (*Pan troglodytes*) along with bonobos (*Pan paniscus*) represent the closest extant relatives of humans sharing a common ancestor approximately 7–8 million years ago (*Langergraber et al., 2012*). Experimental and observational studies, in both the field and in captivity, have documented a range of cognitive abilities that are shared with humans such as tool use and manufacturing (*Shumaker et al., 2011*), symbolic thought (*de and Frans, 1996*), mirror self-

recognition (*Anderson and Gallup, 2015*; *Hecht et al., 2017*) and some basic elements of language (*Savage-Rumbaugh, 1986*; *Savage-Rumbaugh and Lewin, 1994*; *Tomasello and Call, 1997*) like conceptual metaphorical mapping (*Dahl and Adachi, 2013*). This cognitive complexity together with similar neuroanatomical features (*Zilles et al., 1989*; *Rilling and Insel, 1999*; *Gómez-Robles et al., 2013*; *Hopkins et al., 2014*; *Hopkins et al., 2017*) and genetic proximity (*Waterson et al., 2005*) renders these species unique among non-human primates to study the evolutionary origins of the human condition. In view of evolutionary neurobiology, the relatively recent divergence between humans and chimpanzees explains the striking similarities in major gyri and sulci, despite profound differences in overall brain size. Numerous studies using magnetic resonance imaging (MRI) have compared relative brain size, shape, and gyrification in humans and chimpanzees (*Zilles et al., 1989*; *Rilling and Insel, 1999*; *Gómez-Robles et al., 2013*; *Hopkins et al., 2014*; *Hopkins et al., 2017*).

Previous studies of brain aging in chimpanzees have reported minimal indications of atrophy (*Herndon et al., 1999*; *Sherwood et al., 2011*; *Chen et al., 2013*; *Autrey et al., 2014*). Neverthe-less, *Edler et al., 2017* recently found that brains of older chimpanzees' exhibit both neurofibrillary tangles and amyloid plaques, the classical features of Alzheimer's disease (AD). Neurodegeneration in the aging human brain includes marked atrophy in frontal and temporal lobes and decline in glucose metabolism even in the absence of detectable amyloid beta deposition, which increases the likelihood of cognitive decline and development of AD (*Jagust, 2018*). Given the strong association of brain atrophy and amyloid beta in humans, this phenomenon requires further investigation in chimpanzees.

Cortical asymmetry is a prominent feature of brain organization in many primate species (*Hopkins et al., 2015*) and was recently shown in humans in a large-scale ENIGMA (Enhancing Neuroimaging Genetics through Meta-Analysis) study (*Kong et al., 2018*). For chimpanzees, various studies have reported population-level asymmetries in different parts of the brain associated with higher order cognitive functions like tool-use (*Freeman et al., 2004*; *Hopkins et al., 2008*; *Hopkins et al., 2017*; *Hopkins and Nir, 2010*; *Lyn et al., 2011*; *Bogart et al., 2012*; *Gilissen and Hopkins, 2013*) but these results are difficult to compare within and across species, due to the lack of standardized registration and parcellation techniques as found for humans.

To date, there is no common reference space for the chimpanzee brain available to reliably associate and quantitatively compare neuro-anatomical evidence, nor is there a standardized image processing protocol for T1-weighted (T1w) brain images from chimpanzees that matches human imaging standards. With the introduction of voxel-based morphometry (*Ashburner and Friston, 2000*) and the ICBM (international consortium of brain mapping) standard human reference brain templates almost two decades ago (*Mazziotta et al., 2001*), MRI analyses became directly comparable and generally reproducible. In this study, we adapt state-of-the-art MRI (magnetic resonance imaging) processing methods to assess brain aging and cortical asymmetry in the chimpanzee brain. To make this possible, we rely on the largest openly available resource of chimpanzee MRI data: the *National Chimpanzee Brain Resource* (NCBR, http://www.chimpanzeebrain.org/), including in vivo MRI images of 223 subjects from 9 to 54 years of age (Mean age = 26.9 ± 10.2 years). The aim of this study is the creation of a chimpanzee template permitting automated and reproducible image registration, normalization, statistical analysis, and visualization to systematically investigate brain aging and hemispheric asymmetry in chimpanzees.

## Results

Initially, we created the population-based Juna.Chimp (Forschungszentrum *Ju*elich - University Je*na*) T1-template, tissue probability maps (TPM) for tissue classification and a non-linear spatial registration 'Shooting' templates (*Figure 1*) in an iterative fashion at 1 mm spatial resolution. The preprocessing pipeline and templates creation were established using the freely available *Statistical Parametric Mapping* (SPM12 v7487, http://www.fil.ion.ucl.ac.uk/spm/) software and *Computational Anatomy Toolbox* (CAT12 r1704 http://www.neuro.uni-jena.de/cat/). Juna.Chimp templates, the Davi130 parcellation, gray matter (GM) masks utilized, as well as statistical maps from our analysis are interactively accessible and downloadable via the Juna.Chimp web viewer (http://junachimp. inm7.de/).

To enable more direct comparison to previous research, we manually created the Davi130 parcellation (by R.D. and S.V.), a whole brain macroanatomical annotation based on the Juna T1 template

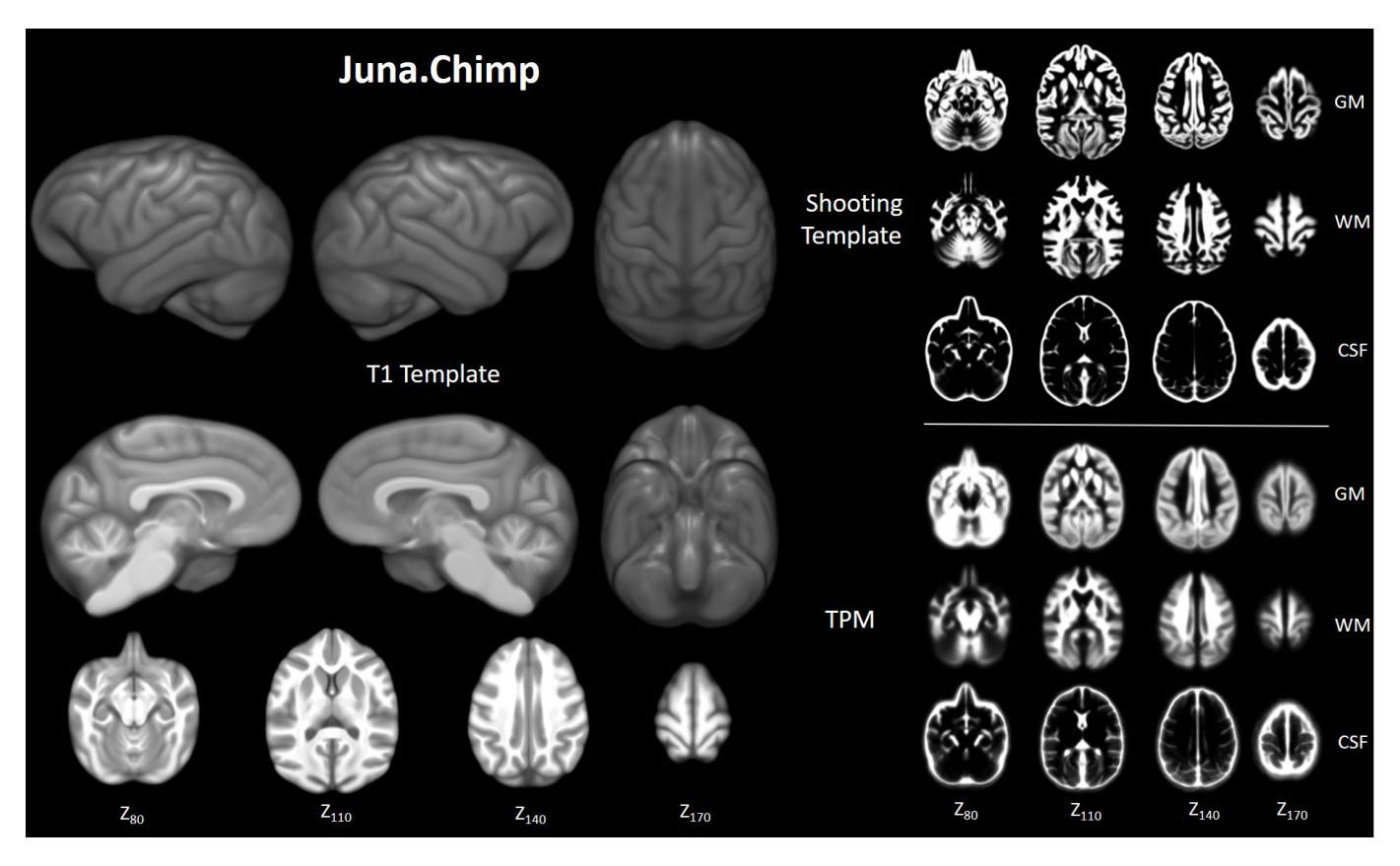

**Figure 1.** Juna.Chimp templates including the average. T1- template, tissue probability maps (TPM), and Geodesic Shooting template. For Shooting templates and TPM axial slices are shown of gray matter (GM), white matter (WM), and cerebrospinal fluid (CSF). All templates are presented at 0.5 mm resolution.

(*Figure 2*). The delineation of regions within the cortex was determined by following major gyri and sulci, whereby, large regions were arbitrarily split into two to three sub-regions of approximate equal size even though histological studies show that micro-anatomical borders between brain regions are rarely situated at the fundus (*Sherwood et al., 2003*; *Schenker et al., 2010*; *Spocter et al., 2010*; *Amunts and Zilles, 2015*). This process yielded 65 regions per hemisphere for a total of 130 regions for the Davi130 macro-anatomical manual parcellation (*Figure 2* and *Figure 2—source data 1*).

Following successful CAT12 preprocessing, rigorous quality control (QC) was employed to identify individual MRI scans suitable for statistical analysis of brain aging and hemispheric asymmetry in chimpanzees. Our final sample consists of 194 chimpanzees including 130 females with an age range of 9–54 years and a mean age of 26.3 ± 9.9 years (*Figure 3A*). The linear regression model with GM fraction of total intracranial volume as the dependent variable and age, scanner field strength, sex, and rearing environment revealed a significant negative association between age and GM (p<0.0001) demonstrating age-related decline in overall GM density (*Figure 3B*). Both sex (p=0.004) and scanner field strength (p<0.0001) showed a significant effect on total GM volume. Therefore, the sample was split into male and female subjects and into 1.5T and 3T scanner, whereby, all sub-samples showed a significant age effect on GM (male: $R^2 = 0.17$, p=0.0004; female $R^2 = 0.13$, p<0.0001, 1.5T: $R^2 = 0.19$, p<0.0001; 3T: $R^2 = 0.09$, p=0.004). There were no significant sex differences of GM decline (p=0.3). The same analysis was conducted on a matched human sample from the IXI dataset (*Figure 3C*; https://brain-development.org/ixi-dataset/). The human sample was matched based on age, sex, and scanner field strength (n = 194, 128 females, 20–78 y/o, mean = 39.4 ± 14.0). As life span and aging processes are different between species, the human sample was matched to chimpanzees roughly by using a factor of 1.5* for age. A significant age-

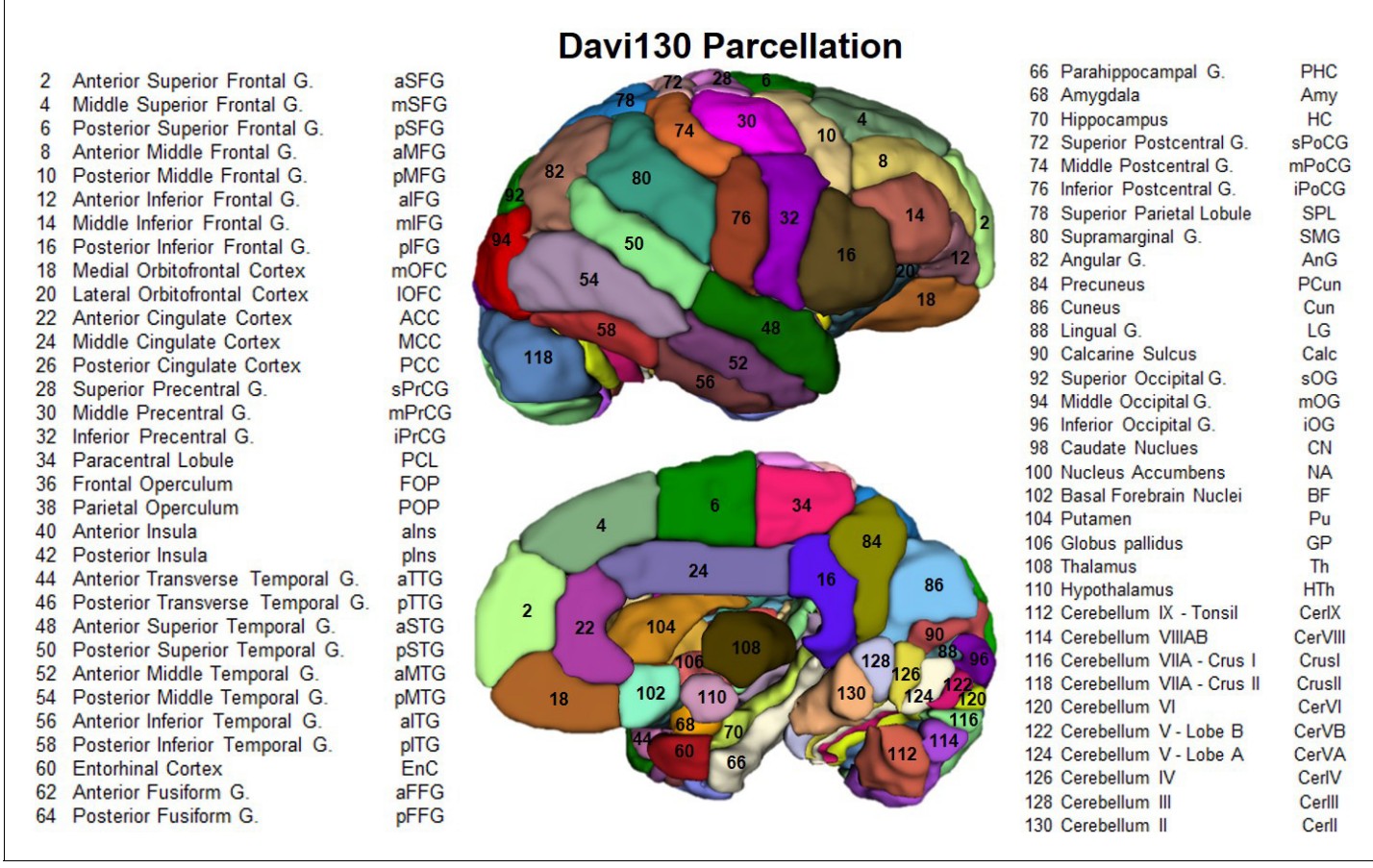

**Figure 2.** Lateral and medial aspect of the Davi130 parcellation right hemisphere. Visible regions are numbered with Davi130 parcellation region numbers and correspond to names in the figure. Even numbers correspond to regions in the right hemisphere (as shown in the figure), while left hemisphere regions are odd numbers. A list of all Davi130 labels can be found at *Figure 2—source data 1*.

The online version of this article includes the following source data for figure 2:

**Source data 1.** Source file for Complete List of Davi130 Labels.

related decline in overall GM (p<0.0001) as well as a significant sex effect (p<0.0001) was also found in the human sample (*Figure 3D*). Similar to the chimpanzee sample, both males and female subjects show a significant age effect on total GM (male: $R^2$ = 0.58, p<0.0001; female: $R^2$ = 0.61, p<0.0001) but with no significant sex differences on GM decline (p=0.8). Although both species present a significant age-related GM decline, humans show a higher negative correlation between age and GM (chimpanzee: $R^2$ = 0.12; human: $R^2$ = 0.55) with less variance as compared to chimpanzees.

Region-based morphometry analysis was applied to test for local effect of age on GM. Linear regression analyses identified 55 of 130 brain regions in the Davi130 parcellation across both hemispheres that were significantly associated with age after family-wise error (FWE) correction for multiple testing (*Figure 4* and *Figure 4—source data 1*). Specifically, GM decline with age was found bilaterally in the superior frontal gyrus (SFG), posterior middle frontal gyrus (pMFG), posterior inferior frontal gyrus (pIFG), lateral orbitofrontal cortex (lOFC), middle and inferior precentral gyrus (PrCG), cingulate gyrus (ACC, MCC, PCC), posterior superior temporal gyrus (pSTG), anterior middle temporal gyrus (aMTG), precuneus (PCun), and lingual gyrus (LG) as well as unilaterally in the right anterior insula (aIns) and middle inferior frontal gyrus (mIFG), in addition to the left superior precentral gyrus (sPrCG), anterior transverse temporal gyrus (aTTG), posterior transverse temporal gyrus (pTTG), paracentral lobule (PCL) and the area around the calcarine sulcus (Calc) within the cerebral cortex. Subcortically, age-related GM decline was found in the bilateral putamen (Pu), caudate nucleus (CN), and the nucleus accumbens (NA), as well as in the superior cerebellum (CerVI, CerIV,

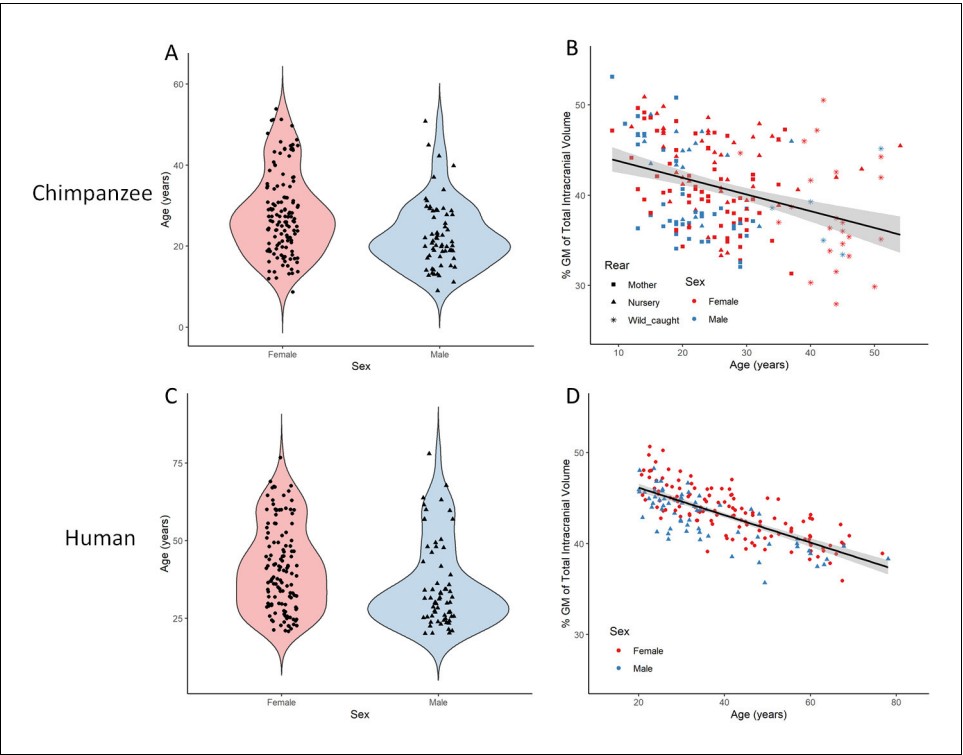

**Figure 3.** Total gray matter volume decline during aging in chimpanzees and matched human sample.
(**A**) Distribution of age and sex in the final sample of 194 chimpanzees. (**B**) Linear relationship between GM and age with standard error for chimpanzee sample. (**C**) Distribution of age and sex in the human (IXI) matched sample of 194 humans. (**D**) Linear relationship between GM and age with standard error for human sample. *Figure 3—figure supplement 1* presents the age and sex distribution of the whole sample (n = 223). *Figure 3—figure supplement 2* presents the age and sex distribution of the whole IXI sample (n = 496).

The online version of this article includes the following figure supplement(s) for figure 3:

**Figure supplement 1.** Age and sex distribution of complete chimpanzee (n = 223) sample separated by scanner field strength.

**Figure supplement 2.** Age and sex distribution of complete IXI human sample (n = 496) separated by scanner field strength.

CerVA, Cer VB, CerIV and right CrusII). Finally, to test for more fine grained effects of aging independently of our macroanatomical parcellation, the same sample was analyzed with VBM revealing additional clusters of GM that are significantly affected by age in chimpanzees (*Figure 5*) after FWE correction using threshold-free cluster enhancement (TFCE) (*Smith and Nichols, 2009*). On top of the regions identified by region-wise morphometry, we found extensive voxel-wise effects throughout the orbitofrontal cortex (OFC), inferior temporal gyrus (ITG), transverse temporal gyrus (TTG), frontal operculum (FOP), parietal operculum (POP), postcentral gyrus (PoCG), supramarginal gyrus (SMG), angular gyrus (AnG), and in parts of the superior parietal lobule (SPL), superior occipital gyrus (sOG), and in inferior parts of the cerebellum.

Hemispheric asymmetry of the chimpanzee brain was assessed for each cortical Davi130 region with a total of 68% (44/65) exhibiting significant cortical asymmetry after FWE correction (*Figure 6* and *Figure 6—source data 1*). The majority of regions were found with greater GM volume in the right hemisphere (n = 32) as compared to the left (n = 12). In the left hemisphere, we found more GM in the SFG, pMFG, insula, anterior TTG, and PCun within the cortex. Rightward cortical asymmetry was located in the anterior MFG, middle and posterior IFG, medial OFC, cingulate gyrus, amygdala, STG, MTG, posterior TTG, anterior and posterior fusiform gyrus (FFG), FOP, POP, middle PrCG, middle and inferior PoCG, SMG, AnG, Calc, as well as the middle occipital gyrus. Within the basal ganglia, leftward GM asymmetry was observed in the Pu, nucleus accumbens (NA), basal forebrain nucleus (BF), and globus pallidus (GP), while, rightward asymmetry in the caudate

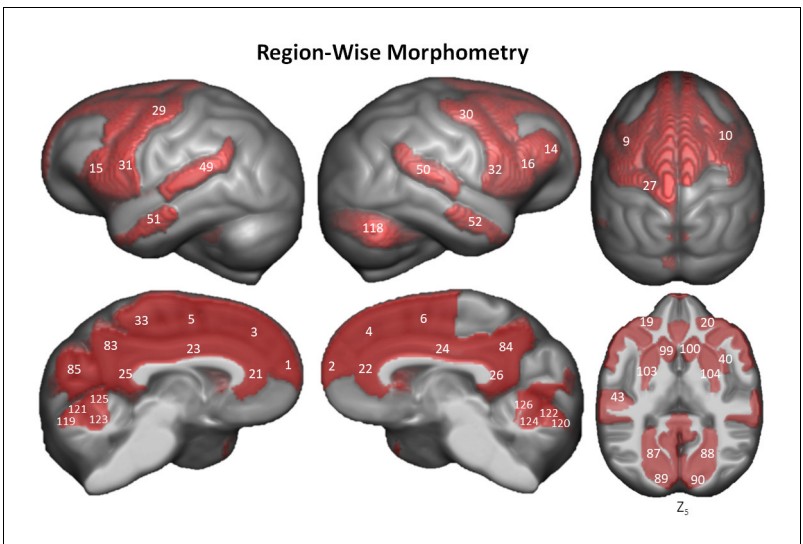

**Figure 4.** Region-wise morphometry in the Davi130 parcellation age regression. Red regions represent Davi130 regions that remained significant at p≤0.05 following FWE correction (Holm method). The T-statistic and p-value for all Davi130 labels can be found in *Figure 4—source data 1*. 1 and 2 – aSFG, 3 and 4 – mSFG, 5 and 6 – pSFG, 9 and 10 – pMFG, 14 – mIFG, 15 and 16 – pIFG, 19 and 20 – lOFC, 21 and 22 – ACC, 23 and 24 – MCC, 25 and 26 – PCC, 27 – sPrCG, 29 and 30 – mPrCG, 31 and 32 – iPrCG, 33 – PCL, 40 – aIns, 43 – aTTG, 49 and 50 – pSTG, 51 and 52 – aMTG, 83 and 84 – PCun, 85 – Cun, 87 and 88 – LG, 89 and 90 – Calc, 97 and 98 – CN, 99 and 100 – NA, 103 and 104 – Pu, 118 – CrusII, 119 and 120 – CerVI, 121 and 122 – CerVB, 123 and 124 – CerVB, 125 and 126 – CerIV.

The online version of this article includes the following source data for figure 4:

**Source data 1.** Aging effect on gray matter in complete Davi130 Labels.

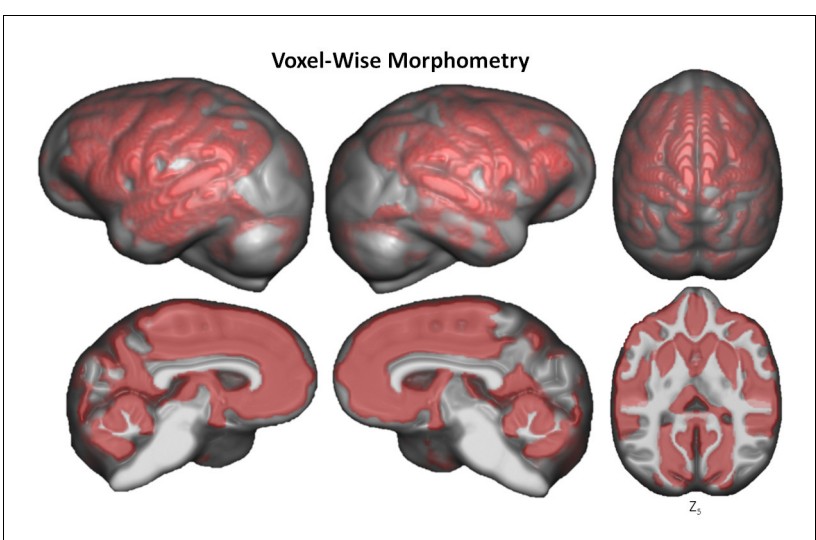

**Figure 5.** Voxel-based morphometry of aging on GM volume. The significant clusters are found using TFCE with FWE correction at p≤0.05.

The online version of this article includes the following figure supplement(s) for figure 5:

**Figure supplement 1.** Voxel-based morphometry of aging on GM volume using TFCE with FWE correction at p≤0.05 without rearing as a covariate.

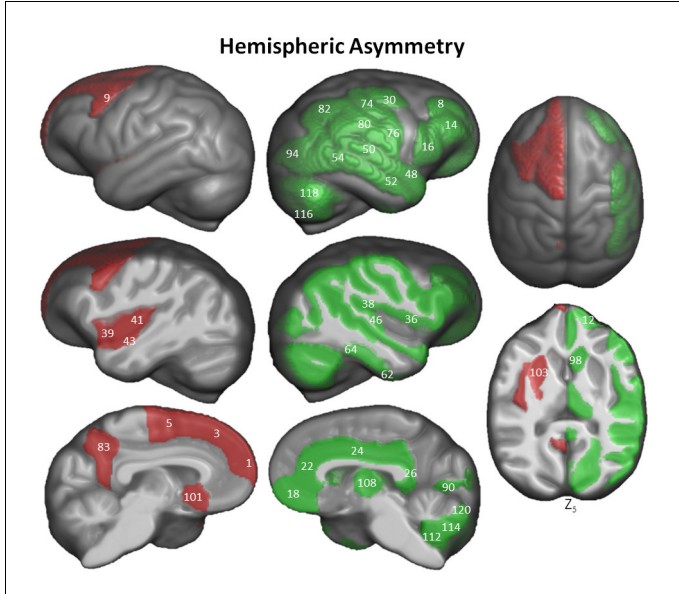

**Figure 6.** Hemispheric asymmetry of Davi130 regions within the chimpanzee sample. Significant leftward (red) and rightward (green) asymmetrical regions are those with a p≤0.05 after FWE correction. The T-statistic and p-value for all Davi130 labels can be found in *Figure 6—source data 1*. 1 – aSFG, 3 – mSFG, 5 – pSFG, 8 – aMFG, 9 – pMFG, 12 – aIFG, 14 – mIFG, 16 – pIFG, 18 – mOFC, 22 – ACC, 24 – MCC, 26 – PCC, 30 – mPrCG, 36 – FOP, 38 – POP, 39 – aIns, 41 – pIns, 43 – aTTG, 46 – pTTG, 48 – aSTG, 50 – pSTG, 52 – aMTG, 54 – pMTG, 62 – aFFG, 64 – pFFG, 74 – mPoCG, 76 – iPoCG, 80 – SMG, 82, AnG, 83 – PCun, 90 – Calc, 94 – mOG, 98 – CN, 101 – BF, 103 – Pu, 108 – Th, 112 – CerIX, 114 – CerVIII, 115 –CrusI, 118 –CrusII, 120 – CerVI.

The online version of this article includes the following source data for figure 6:

**Source data 1.** Complete Davi130 labels hemispheric asymmetry.

nucleus (CN) and thalamus (Th). The cerebellum exclusively showed rightward GM asymmetry in the posterior cerebellar lobe (CerIX, CerVIII, CrusI, CrusII, CerVI). The hemispheric asymmetry did not show a decipherable pattern.

## Discussion

As a common reference space for the analysis of chimpanzee brain data, we created the Juna.Chimp template, constructed from a large heterogeneous sample of T1w MRI's from the NCBR. The Juna. Chimp template includes a reference T1-template, along with probability maps of brain and head tissues accompanied by a Geodesic Shooting template for the publicly available SPM12/CAT12 pre-processing pipeline to efficiently segment and accurately spatially normalize individual chimpanzee T1w images. The T1-template and TPM can also be used as the target for image registration with other popular software packages, such as FSL (https://fsl.fmrib.ox.ac.uk/fsl) or ANTs (http://stnava. github.io/ANTs/). Furthermore, our processing pipeline and templates can be utilized for data-driven approaches to create connectivity-based and structural covariance parcellations of the chimpanzee brain (*Alexander-Bloch et al., 2013*; *Eickhoff et al., 2015*).

Additionally, we provide the manually segmented, macro-anatomical Davi130 whole-brain parcellation comprising 130 cortical, sub-cortical and cerebellar brain regions, which enables systematic extraction of volumes-of-interest from chimpanzee MRI data. The image processing pipeline and Davi130 parcellation were used to investigate ageing and interhemispheric asymmetry in the chimpanzee brain. The Davi130 parcellation was realized by utilizing macroscopic gyral and sulcal features such as peaks, fundi, and bends to represent the chimpanzee brain in a reduced dimensional space based on macro-anatomical landmarks. Despite the evidence that true microanatomical borders between brain areas rarely coincide with macro-anatomical patterns (*Amunts and Zilles, 2015*), macroscopic brain parcellations like Desikan-Killany human atlas (*Desikan et al., 2006*) have successfully been utilized in many studies furthering our understanding of brain structure, function, and

disease (*Kong et al., 2020*; *van den Heuvel et al., 2020*). The Davi130 parcellation of the chimpanzee brain serves two main purposes. First, regions in Juna.Chimp template space enable increased interpretability and reproducibility of morphometric analyses and comparability between studies, even retrospectively. Second, our manual subdivision reduces the statistical problem of multiple testing for mass univariate approaches like VBM to uncover subtle brain - behavior relationships. Furthermore, our macroscopic parcellation mitigates the curse of dimensionality for multivariate machine learning methods to be applied to the relatively small samples like the NCBR.

We found clear evidence of global and local GM decline in the aging chimpanzee brain even though previous research into age-related changes in chimpanzee brain organization has shown little to no effect (*Herndon et al., 1999*; *Sherwood et al., 2011*; *Chen et al., 2013*; *Autrey et al., 2014*). (*Herndon et al., 1999*; *Sherwood et al., 2011*; *Chen et al., 2013*; *Autrey et al., 2014*). This can be attributed on the one hand to the larger number of MRI scans available via the NCBR including 30% of older subjects with 55 individuals over 30 and 12 over 45 years of age, which is crucial for modelling the effect of aging (*Chen et al., 2013*; *Autrey et al., 2014*). On the other hand, state-of-the-art image processing enabled the creation of the species-specific Juna.Chimp templates, which largely improves tissue segmentation and registration accuracy (*Ashburner and Friston, 2000*). Non-linear registration was also improved by the large heterogeneous sample utilized for the creation of the templates encompassing a representative amount of inter-individual variation. We used the well-established structural brain imaging toolbox CAT12 to build a reusable chimpanzee preprocessing pipeline catered towards analyzing local tissue-specific anatomical variations as measured with T1w MRI. The Davi130-based region-wise and the voxel-wise morphometry analysis consistently showed localized GM decline in lateral frontal cortex, lOFC, precentral gyrus, cingulate gyrus, PCun, medial parietal and occipital cortex, the basal ganglia, and superior cerebellum. The VBM approach additionally produced evidence for age effects in bilateral mOFC, PoCG, inferior temporal regions, inferior and superior lateral parietal cortex, sOG, and throughout the cerebellum. These additional effects can be expected, as VBM is more sensitive to GM changes due to aging (*Kennedy et al., 2009*). The multiple brain regions revealing GM decline reported here in both approaches have also been shown to exhibit GM atrophy during healthy aging in humans (*Good et al., 2001b*; *Kennedy et al., 2009*; *Crivello et al., 2014*; *Minkova et al., 2017*). Additionally, there was no significant difference in the age-related decline between humans and chimpanzee (*Figure 3*), even though a larger negative correlation with less variance was found in the matched human sample, which demonstrates a commonality in the healthy aging process of chimpanzees that was thought to be specific to humans. In general, GM atrophy in chimpanzees occurs across the entire cortex, sub-cortical regions, and cerebellum, however, certain local areas decline at a relative extended rate within the frontal, temporal, and parietal lobes (*Fjell et al., 2014*). Several Davi130 regions within the frontal lobe (SFG, MFG, IFG, and lOFC) have been previously reported in corresponding human loci in relation to GM volume decline due to aging (*Kennedy et al., 2009*; *Crivello et al., 2014*; *Fjell et al., 2014*; *Minkova et al., 2017*). Furthermore, aging effects in temporal (STG) and parietal (PoCG, AnG, PCun) regions in chimpanzees have additionally been revealed in analogous human areas (*Good et al., 2001b*; *Kennedy et al., 2009*; *Crivello et al., 2014*; *Fjell et al., 2014*; *Minkova et al., 2017*). The same is true for the superior occipital gyrus and caudate nucleus (*Crivello et al., 2014*). Similar age-related total GM decline along with presentation in homologous brain areas suggests common underlying neurophysiological processes in humans and chimpanzees due to shared primate evolution.

Very recently, it has been shown that stress hormone levels increase with age in chimpanzees, a process previously thought to only occur in humans, which can cause GM volume decline (*Emery Thompson et al., 2020*). This further strengthens the argument that age-related GM decline is also shared by humans closest relative, the chimpanzee. Furthermore, *Edler et al., 2017* found Alzheimer's disease-like accumulation of amyloid beta plaques and neurofibrillary tangles located predominantly in prefrontal and temporal cortices in a sample of elderly chimpanzees between 37 and 62 years of age. As the aggregation of these proteins is associated with localized neuronal loss and cortical atrophy in humans (*La Joie et al., 2012*; *Lladó et al., 2018*), the age-related decline in GM volume shown here is well in line with the findings by *Jagust, 2016* associating GM atrophy with amyloid beta. These findings provide a biological mechanism for accelerated GM decrease in prefrontal, limbic, and temporal cortices found in chimpanzees. In contrast, elderly rhesus monkeys show GM volume decline without the presence of neurofibrillary tangles (*Alexander et al., 2008*;

*Shamy et al., 2011*). Taken together, regionally specific GM atrophy seems to be a common aspect of the primate brain aging pattern observed in macaque monkeys, chimpanzees, and humans. To make a case for the existence of Alzheimer's disease in chimpanzees, validated cognitive tests for Alzheimer's-like cognitive decline in non-human primates are needed, to test for direct associations between cognitive decline with tau pathology and brain atrophy.

To further analyze the possible moderator effects on aging, we considered the historical composition of the NCBR sample, with respect to the rearing environment. The majority of elderly chimpanzees over 40 years old (23/26) were born in the wild and captured at a young age, whereas only very few chimpanzees under 40 were wild born (5/168). The capture, separation from their mothers, and subsequent transport to the research centers can be considered a traumatic event with possible lasting effects on brain development and morphology (*Bremner, 2006*). In captivity, different chimpanzee-rearing experiences, either by their mother or in a nursery, has been shown to affect brain morphology (*Bogart et al., 2014*; *Bard and Hopkins, 2018*). The same should be expected in comparison of captive and wild-born chimpanzees. The disproportionate distribution of rearing and early life experiences likely influences our cross-sectional analyses of the effect of aging on GM volume. However, we have some reason to be confident that the aging effect shown here is not solely driven by these factors as rearing environment was added as a covariate to all age regression models and the VBM age regression model with and without rearing as a covariate are almost identical (*Figure 5* and *Figure 5—figure supplement 1* respectively). Moreover, the GM decline we found is extensive, widespread, and also present in chimpanzees under 30 years of age (p<0.0001), where 99% are captive born (143/144).

Hemispheric asymmetry was found in 68% (44/65) of all regions of the Davi130 parcellation, reproducing several regional findings reported in previous studies using diverse image processing methods as well as uncovering numerous novel population-level asymmetries. Previous studies utilizing a region-wise approach based on hand-drawn or atlas derived regions to analyze asymmetry in cortical thickness also reported leftward asymmetry of the insula (*Hopkins et al., 2017*) and rightward lateralization of cortical thickness of the PCC (*Hopkins et al., 2017*) as well as STG, MTG, and SMG (*Hopkins and Avants, 2013*). Previous VBM findings also revealed leftward asymmetry in the anterior SFG (*Hopkins et al., 2008*) along with rightward lateralization of the MFG, PrCG, PoCG, mOG, and CrusII (*Hopkins et al., 2008*; *Hopkins and Avants, 2013*). In the current study, new regions of larger GM volume in the left hemisphere were found in frontal (pMFG, mSFG, pSFG), temporal (aTTG), and parietal (PCun) cortices as well as in the basal ganglia (BF, GP, Pu). Novel rightward asymmetries could also be seen in the frontal (IFG, mOFC, FrOP), limbic (CC, Amy), temporal (pTTG, FFG), parietal (POP, AnG), and occipital (Calc) cortices besides the basal ganglia (Th, CN) and the cerebellum (CrusI, CerIX, CerVI, CerVIII).

The Davi130s' region pTTG which contains the planum temporale (PT), presented significant rightward lateralization, while previous studies of the PT have shown leftward asymmetry in chimpanzee GM volume, surface area (*Hopkins and Nir, 2010*), and cytoarchitecture (*Zilles et al., 1996*; *Gannon et al., 1998*; *Spocter et al., 2010*). A possible reason for the divergence in this finding is that the anterior border of PT (*Hopkins and Nir, 2010*) lies several millimeters posterior from the anterior posterior split of the Davi130 TTG. Additionally, the left lateral sulcus in the Juna.Chimp template appears to proceed further posteriorly and superiorly compared to the right, which is consistent with previous findings in asymmetrical length of the POP in chimpanzees (*Gilissen and Hopkins, 2013*) and Sylvian fissure length in old world monkeys (*Lyn et al., 2011*; *Marie et al., 2018*). Population-level asymmetries in the pIFG in chimpanzees were documented almost two decades ago by *Cantalupo and Hopkins, 2001*, who reported a leftward asymmetry in pIFG volume in a small sample of great apes. In subsequent studies, this result could not be replicated when considering GM volume (*Hopkins et al., 2008*; *Keller et al., 2009*) or cytoarchitecture (*Schenker et al., 2010*). We also failed to find a leftward asymmetry in GM volume for the pIFG, in contrary to asymmetries found in humans (*Amunts et al., 1999*; *Uylings et al., 2006*; *Keller et al., 2009*).

A substantial amount of regions presenting significant inter-hemispheric differences of local morphology in chimpanzees has also been shown in humans (*Good et al., 2001a*; *Plessen et al., 2014*; *Kong et al., 2018*). Specifically, leftward lateralization has been found in human analogous regions of the SFG and insula utilizing GM thickness and volume in voxel-wise and atlas derived region-wise approaches (*Good et al., 2001a*; *Takao et al., 2011*; *Plessen et al., 2014*; *Kong et al., 2018*). Rightward asymmetry in the Davi130 regions IFG, STG, MTG, AnG, mOG, Calc, in addition to the

thalamus and lateral cerebellum is documented in the human brain also using both VBM and surface measures (*Good et al., 2001a*; *Takao et al., 2011*; *Plessen et al., 2014*; *Kong et al., 2018*). Gross hemispheric asymmetry in humans follows a general structure of frontal rightward and occipital leftward asymmetry known as the 'Yakovlevian torque' (*Toga and Thompson, 2003*). This general organizational pattern of asymmetry was not apparent in the chimpanzee (*Li et al., 2018*).

The NCBR offers the largest and richest openly available dataset of chimpanzee brain MRI scans acquired over a decade with 1.5T and 3T MRI at two locations, capturing valuable inter-individual variation in one large heterogeneous sample. To account for the scanner effect on GM estimation, field strength was modeled as a covariate of no interest for analyzing the age effect on GM volume. The focus of this study was the analysis of GM volume, even though the CAT12 image processing pipeline enables surface projection and analysis. Consequently, the next step will be the application of CAT12 to analyze cortical surface area, curvature, gyrification, and thickness of the chimpanzee brain, to include behavioral data and the quantitative comparison to humans and other species, as cortical surface projection permits a direct inter-species comparison due to cross-species registration.

## Conclusion

In conclusion, we present the new chimpanzee reference template Juna.Chimp, TPM's, the Davi130 whole-brain parcellation, and the CAT12 preprocessing pipeline which is ready-to-use by the wider neuroimaging community. Investigations of age-related GM changes in chimpanzees using both region-wise and voxel-based morphometry showed substantial atrophy with age, which was also apparent in a matched human sample providing further evidence for human-like physiological aging processes in the chimpanzee brain. Examining population-based hemispheric asymmetry in chimpanzees showed a general rightward lateralization of higher GM volume without the presence of a distinct pattern like the 'Yakovlevian torque' seen in humans.

# Materials and methods

**Key resources table**

| Reagent type (species) or resource | Designation | Source or reference | Identifiers | Additional information |
|---|---|---|---|---|
| Software, algorithm | CAT12 | http://www.neuro.uni-jena.de/cat/ | RRID:SCR_019184 | |
| Software, algorithm | NCBR | http://www.chimpanzeebrain.org/ | RRID:SCR_019183 | |
| Software, algorithm | MATLAB | http://www.mathworks.com/products/matlab/ | RRID:SCR_001622 | |
| Software, algorithm | SPM | http://www.fil.ion.ucl.ac.uk/spm/ | RRID:SCR_007037 | |
| Software, algorithm | RStudio | http://www.rstudio.com/ | RRID:SCR_000432 | |
| Software, algorithm | 3D Slicer | http://slicer.org/ | RRID:SCR_005619 | |

## Subject information and image collection procedure

This study analyzed structural T1w MRI scans of 223 chimpanzees (137 females; 9–54 y/o, mean age 26.9 ± 10.2 years, *Figure 3—figure supplement 1*) from the NCBR (http://www.chimpanzeebrain.org/). The chimpanzees were housed at two locations including, the *National Center for Chimpanzee Care* of *The University of Texas MD Anderson Cancer Center* (UTMDACC) and the *Yerkes National Primate Research Center* (YNPRC) of Emory University. The standard MR imaging procedures for chimpanzees at the YNPRC and UTMDACC are designed to minimize stress for the subjects. For an

in-depth explanation of the imaging procedure please refer to *Autrey et al., 2014*. Seventy-six chimpanzees were scanned with a Siemens Trio 3 Tesla scanner (Siemens Medical Solutions USA, Inc, Malvern, Pennsylvania, USA). Most T1w images were collected using a three-dimensional gradient echo sequence with $0.6 \times 0.6 \times 0.6$ resolution (pulse repetition = 2300 ms, echo time = 4.4 ms, number of signals averaged = 3). The remaining 147 chimpanzees were scanned using a 1.5T GE echo-speed Horizon LX MR scanner (GE Medical Systems, Milwaukee, WI), predominantly applying gradient echo sequence with $0.7 \times 0.7 \times 1.2$ resolution (pulse repetition = 19.0 ms, echo time = 8.5 ms, number of signals averaged = 8).

## DICOM conversion and de-noising

The structural T1w images were provided by the NCBR in their original DICOM format and converted into Nifti using MRIcron (*Rorden and Brett, 2000*). If multiple scans were available, the average was computed. Following DICOM conversion, each image was cleaned of noise (*Manjón et al., 2010*) and signal inhomogeneity and resliced to 0.6 mm isotropic resolution. Finally, the anterior commissure was manually set as the center (0,0,0) of all Nifti's to aid in affine preprocessing.

## CAT12 preprocessing segmentation

Structural image segmentation in CAT12 builds on the TPM-based approach employed by SPM12, whereby, the gray/white image intensity is aided with a priori tissue probabilities in initial segmentation and affine registration as it is in common template space. Another advantage of a TPM is that one has a template for initial affine registration, which then enables the segment maps to be non-linearly registered and spatially normalized to corresponding segment maps of the chimpanzee shooting templates. Lowering the possibility for registration errors improves the quality of the final normalized image. Improving upon SPM's segmentation (*Ashburner and Friston, 2005*), CAT12 employs Local Adaptive Segmentation (LAS) (*Dahnke et al., 2012*), Adaptive Maximum A Posterior segmentation(AMAP) (*Dahnke and Gaser, 2017*; *Gaser et al., 2020*), and Partial Volume Estimation (PVE) (*Tohka et al., 2004*). LAS creates local intensity transformations for all tissue types to limit GM misclassification due to varying GM intensity in regions such as the occipital, basal ganglia, and motor cortex because of anatomical properties (e.g. high myelination and iron content). AMAP segmentation takes the initially segmented, aligned, and skull stripped image created utilizing the TPM and disregards the a priori information of the TPM, to conduct an adaptive AMAP estimation where local variations are modeled by slowly varying spatial functions (*Rajapakse et al., 1997*). Along with the classical three tissue types for segmentation (GM, WM, and CSF) based on the AMAP estimation, an additional two PVE classes (GM-WM and GM-CSF) are created resulting in an estimate of the fraction of each tissue type contained in each voxel. These features outlined above of our pipeline allow for more accurate tissue segmentation and therefore a better representation of macroanatomical GM levels for analysis.

## Creation of chimpanzee templates

An iterative process as by *Franke et al., 2017* was employed to create the Juna.Chimp template, with T1 average, Shooting registration template (*Ashburner and Friston, 2011*), as well as the TPM (*Figure 7*). Initially, a first-generation template was produced using the 'greater_ape' template delivered by CAT (*Franke et al., 2017*; *Gaser et al., 2020*) that utilizes data provided in *Rilling and Insel, 1999*. The final segmentation takes the bias-corrected, intensity-normalized, and skull-stripped image together with the initial SPM-segmentation to conduct an AMAP estimation (*Rajapakse et al., 1997*) with a partial volume model for sub-voxel accuracy (*Tohka et al., 2004*). The affine normalized tissue segments of GM, white matter (WM), and cerebrospinal fluid (CSF) were used to create a new Shooting template that consists of four major non-linear normalization steps allowing to normalize new scans. To create a chimpanzee-specific TPM, we average the different Shooting template steps to benefit from the high spatial resolution of the final Shooting steps but also include the general affine aspects to avoid over-optimization. Besides the brain tissues the TPM also included two head tissues (bones and muscles) and a background class for standard SPM12 (*Ashburner and Friston, 2005*) and CAT12 preprocessing. An internal CAT atlas was written for each subject and mapped to the new chimpanzee template using the information from the Shooting registration. The CAT atlas maps were averaged by a median filter and finally manually

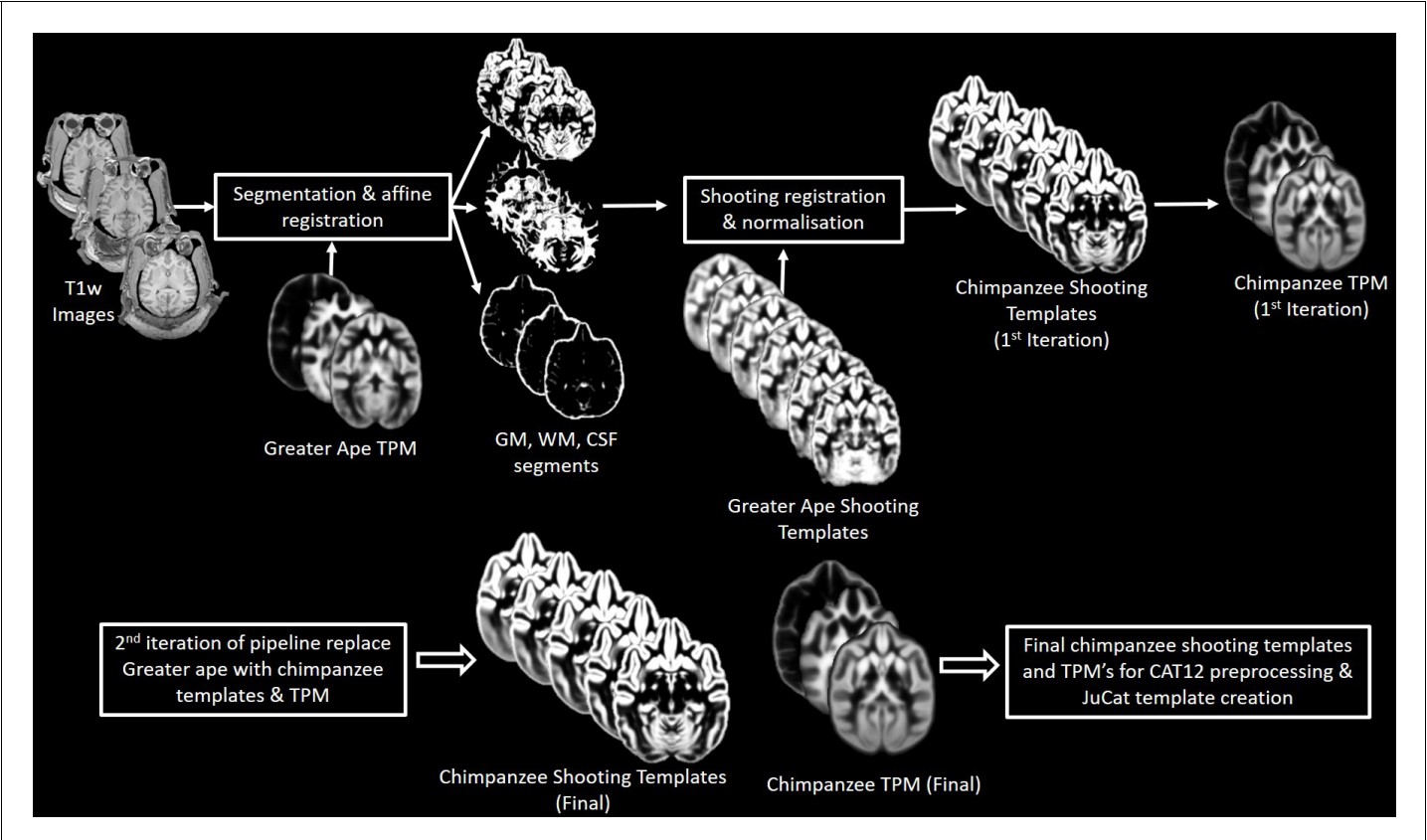

**Figure 7.** Workflow for creation of chimpanzee-specific shooting template and TPM, which can then be used in CAT12 structural preprocessing pipeline to create the Juna.Chimp template. The resulting chimpanzee-shooting template, TPM and CAT atlas establishes the robust and reliable base to segment and spatially normalize the T1w images utilizing CAT12's processing pipeline (*Dahnke and Gaser, 2017*; *Gaser et al., 2020*).

corrected. This initial template was then used in the second iteration of CAT segmentation to establish the final chimpanzee-specific Juna.Chimp template, which was imported into the standard CAT12 preprocessing pipeline to create the final data used for the aging and asymmetry analyses.

## Davi130 parcellation

The average T1 and final Shooting template were used for a manual delineation of macro-anatomical GM structures. Identification and annotation of major brain regions were performed manually using the program, 3D Slicer 4.10.1 (https://www.slicer.org). The labeling enables automated, region-based analysis of the entire chimpanzee brain and allows for robust statistical analysis. Nomenclature and location of regions were ascertained by consulting both chimpanzee and human brain atlases (*Bailey and Bonin GV, 1950*; *Mai et al., 2015*). The labeling was completed by two authors (S.V. and R.D.) and reviewed by two experts of chimpanzee brain anatomy (C.C.S. and W.D.H.). A total of 65 GM structures within the cerebrum and cerebellum of the left hemisphere were annotated and then flipped to the right hemisphere. The flipped annotations were then manually adapted to the morphology of the right hemisphere to have complete coverage of the chimpanzee brain with 130 labels.

The location of macroscopic brain regions was determined based on major gyri of the cerebral cortex, as well as distinct anatomical landmarks of the cerebellar cortex, and basal ganglia. Of note, the border between two adjacent gyri was set as the mid-point of the connecting sulcus, generally at the fundus. Large gyri were further subdivided into two or three parts based on their size and structural features to enable greater spatial resolution and better inter-regional comparison. Naming of regional subdivisions were based on spatial location, for example, anterior, middle, posterior, as

these splits are based on macroanatomical features and do not necessarily correspond to functional parcellations.

Considering the limitations of macroscopic features present in T1w, we utilized distinct morphological representations to split large gyri, such as gyral/sulcal folds and continuation of sulci. If a distinguishable feature could not be determined, rough distance and regional size was employed as border defining criteria. The splits of the lateral temporal lobe, including the TTG, followed a continuation of the inferior portion of the postcentral sulcus that angles slightly posteriorly to better account for the increase in length of the gyri as it proceeds inferiorly. The central sulcus as well as the adjacent pre- and postcentral gyri contain a knob or U-shaped bend proceeding posteriorly. The superior beginning and inferior end of this bend were employed for the two splits of these gyri. Additionally, the central sulcus is the border between the frontal and parietal lobes, therefore, the FOP – POP split occurs at the termination of the central sulcus at the lateral fissure. Within the frontal cortex, the anterior posterior split of the MFG is at the meeting point of the middle frontal sulcus and the superior precentral sulcus, which translates to the inferior bend of the MFG. The tip of the fronto-orbito sulcus was used as an anchor point for the split of the pIFG and mIFG. The middle anterior split of the IFG was then determined by distance, whereby the remaining gyrus was separated into equally sized parts. The cingulate cortex anterior, middle, and posterior subdivisions were delineated by splits following the anterior and posterior bends of the gyrus around the corpus callosum. The cerebellum was divided into its major lobes which are quite similar across primates (*Apps and Hawkes, 2009*). Finally, splits within the OFC, SFG, and insula were based on equal size and/or distance.

## Quality control

CAT12 provides quality measures pertaining to noise, bias inhomogeneities, resolution and an overall compounded score of the original input image. Using these ratings, poor images were flagged for visual inspection when they were two standard deviations (std) away from the sample mean of each rating. The preprocessed modulated GM maps were then tested for sample inhomogeneity separately for each scanner (3T and 1.5T) and those that have a mean correlation below two std were flagged for visual inspection. Once the original image was flagged, affine GM, and modulated GM maps were inspected for poor quality, tissue misclassification, artefacts, irregular deformations, and very high intensities. For the second and third iteration, the passed modulated GM maps were tested again for mean correlation as a complete sample, flagging the images below two std for visual inspection, looking for the same features as in the initial QC iteration. Following the three iterations of QC a total of 194 of 223 chimpanzee MRI's (130 females, 9–54 y/o, mean = 26.2 ± 9.9) qualified for statistical analysis.

## Age-related changes in total gray matter

A linear regression model was used to determine the effect of aging on total GM volume. Firstly, total GM volume for each subject was converted into a percentage of total intracranial volume (TIV) to account for the variation in head size. This was then entered into a linear regression model as the dependent variable with age, sex, scanner field strength, and rearing as the independents. Sex-specific models were conducted with males and females separately using age as the only dependent variable. The slope of each regression line was determined using $R^2$ and a p-value of $p \leq 0.05$ was used to determine the significant effect of age and sex on total GM volume. The IXI brain development dataset (http://brain-development.org/ixi-dataset/) was utilized to compare the age effect on total GM volume between chimpanzees and humans, as it includes subjects with a wide age range and T1w images from MRI scanners of both 1.5T and 3T field strength. Prior to matching the IXI sample to the QC passed chimpanzee sample, all images collected from the Institute of Psychiatry (IOP) were removed to keep similarity to the chimpanzee sample of a single 1.5T scanner. After removing subjects without meta data, a total of 496 subjects (*Figure 3—figure supplement 2*) were used for matching to the chimpanzee sample regarding age, sex, and scanner field strength. To enable age matching between species, a factor of 1.5 of chimpanzee age was used to roughly calculate the comparable human age. This factor was chosen based on the comparable life span of the two species, because a chimpanzee 40+ years is considered elderly and so is a 60+ year old human, also a 60+ year old chimpanzee is very old and uncommon similarly to a human 90+ years old.

Furthermore, the age of sexual maturity in humans is 19.5 years, while in chimps it is 13.5 years which is also approximately a difference of 1.5 (*Robson and Wood, 2008*). The sample matching was conducted using the 'MatchIt' (*Ho et al., 2007*) R package (https://cran.r-project.org/package= MatchIt) and utilizing the 'optimal' (*Hansen and Klopfer, 2006*) algorithm. The matched human sample contained 194 subjects (128 females, 20–78 y/o, mean = 39.4 ± 14.0) for statistical analysis.

## Age-related changes in gray matter using Davi130 parcellation

The Davi130 parcellation was applied to the modulated GM maps to conduct region-wise morphometry analysis. First, the Davi130 regions were masked with a 0.1 GM mask to remove all non-GM portions of the regions. Subsequently, the average GM intensity of each region for all QC-passed chimpanzees was calculated. A multiple regression model was conducted for the labels from both hemispheres, whereby, the dependent variable was GM volume and the predictor variables were age, sex, TIV, scanner strength, and rearing. Significant age-related GM decline was established for a Davi130 label with a $p \leq 0.05$, after correcting for multiple comparisons using FWE (*Holm, 1979*).

## Voxel-based morphometry

VBM analysis was conducted using CAT12 to determine the effect of aging on local GM volume. The modulated and spatially normalized GM segments from each subject were spatially smoothed with a 4 mm FWHM (full width half maximum) kernel prior to analyses. To restrict the overall volume of interest, an implicit 0.4 GM mask was employed. As MRI field strength is known to influence image quality, and consequently, tissue classification, we included scanner strength in our VBM model as a covariate. The dependent variable in the model was age, with covariates of TIV, sex, scanner strength, and rearing. The VBM model was corrected for multiple comparisons using TFCE with 5000 permutations (*Smith and Nichols, 2009*). Significant clusters were determined at $p \leq 0.05$, after correcting for multiple comparisons using FWE.

## Hemispheric asymmetry

As for the age regression analysis, all Davi130 parcels were masked with a 0.1 GM mask to remove non-GM portions within regions. Cortical hemispheric asymmetry of Davi130 labels was determined using the formula *Asym = (L - R) / (L + R) * 0.5* (*Kurth et al., 2015*; *Hopkins et al., 2017*), whereby L and R represent the average GM volume for each region in the left and right hemisphere, respectively. Therefore, the bi-hemispheric Davi130 regions were converted into single *Asym* labels (n = 65) with positive *Asym* values indicating a leftward asymmetry, and negative values, a rightward bias. One-sample *t*-tests were conducted for each region under the null hypothesis of *Asym = 0*, and significant leftward or rightward asymmetry was determined with a $p \leq 0.05$, after correcting for multiple comparisons using FWE (*Holm, 1979*).

## Exemplar pipeline workflow

To illustrate the structural processing pipeline, we have created exemplar MATLAB SPM batch scripts that utilizes the Juna.Chimp templates in CAT12's preprocessing workflow to conduct segmentation, spatial registration, and finally some basic age analysis on an openly available direct-to-download chimpanzee sample (http://www.chimpanzeebrain.org/). These scripts require the appropriate templates which can be downloaded from the Juna.Chimp web viewer (SPM/CAT_templates. zip) and then place the templates_animals/folder into the latest version CAT12 Toolbox directory (CAT12.7 r1609). The processing parameters are similar to those conducted in this study, although different DICOM conversions and denoising were conducted. Further information regarding each parameter can be viewed when opening the script in the SPM batch as well as the provided comments and README file. The code for the workflow in addition to the code used to conduct the aging effect and asymmetry analyses can be found here (https://github.com/viko18/ JunaChimp; *Vickery, 2020*; copy archived at swh:1:rev: 411f0610269416d4ee04eaf9670a9dc84e829ea0).

## Acknowledgements

We thank Jona Fischer for the creation of the interactive Juna.Chimp web viewer adapted from nehuba (github).

## Additional information

### Funding

| Funder | Grant reference number | Author |
|---|---|---|
| Helmholtz Association | Helmholtz Portfolio Theme 'Supercomputing and Modelling for the Human Brain | Sam Vickery Simon B Eickhoff Felix Hoffstaedter |
| Horizon 2020 | 945539 (HBP SGA 3) | Sam Vickery Simon B Eickhoff Felix Hoffstaedter |
| Helmholtz Association | Initiative and Networking Fund | Svenja Caspers |
| Horizon 2020 | 785907 (HBP SGA 2) | Svenja Caspers |
| National Institutes of Health | NS-42867 | William D Hopkins |
| National Institutes of Health | NS092988 | Chet C Sherwood |
| James S. McDonnell Foundation | 220020293 | Chet C Sherwood |
| Inspire Foundation | SMA-1542848 | Chet C Sherwood |
| National Institutes of Health | U42-OD011197 | Steven J Schapiro |
| Deutsche Forschungsgemeinschaft | 417649423 | Robert Dahnke |
| National Institutes of Health | NS-73134 | William D Hopkins |
| National Institutes of Health | NS-92988 | William D Hopkins |

The funders had no role in study design, data collection and interpretation, or the decision to submit the work for publication.

### Author contributions

Sam Vickery, Conceptualization, Data curation, Software, Formal analysis, Investigation, Visualization, Methodology, Writing - original draft, Project administration, Writing - review and editing; William D Hopkins, Conceptualization, Resources, Data curation, Funding acquisition, Validation, Methodology, Writing - original draft, Writing - review and editing; Chet C Sherwood, Resources, Data curation, Funding acquisition, Validation, Methodology, Writing - original draft, Writing - review and editing; Steven J Schapiro, Resources, Data curation, Funding acquisition, Writing - original draft, Writing - review and editing; Robert D Latzman, Conceptualization, Data curation, Writing - original draft, Writing - review and editing; Svenja Caspers, Conceptualization, Supervision, Writing - original draft, Writing - review and editing; Christian Gaser, Resources, Software, Methodology, Writing - original draft, Writing - review and editing; Simon B Eickhoff, Conceptualization, Resources, Supervision, Funding acquisition, Writing - original draft, Project administration, Writing - review and editing; Robert Dahnke, Conceptualization, Resources, Data curation, Software, Supervision, Validation, Investigation, Methodology, Writing - original draft, Project administration, Writing - review and editing; Felix Hoffstaedter, Conceptualization, Software, Supervision, Validation, Methodology, Writing - original draft, Project administration, Writing - review and editing

### Author ORCIDs

Sam Vickery (iD) https://orcid.org/0000-0001-6732-7014
Chet C Sherwood (iD) http://orcid.org/0000-0001-6711-449X
Robert D Latzman (iD) http://orcid.org/0000-0002-1175-8090

Simon B Eickhoff ⬚ http://orcid.org/0000-0001-6363-2759
Felix Hoffstaedter ⬚ https://orcid.org/0000-0001-7163-3110

## Ethics

Animal experimentation: the chimpanzee imaging data were acquired under protocols approved by the Yerkes National Primate Research Center (YNPRC) at Emory University Institutional Animal Care and Use Committee (Approval number YER2001206).

## Decision letter and Author response

Decision letter https://doi.org/10.7554/eLife.60136.sa1
Author response https://doi.org/10.7554/eLife.60136.sa2

## Additional files

### Supplementary files

- Transparent reporting form

### Data availability

The T1-weighted MRI's are available at the National Chimpanzee Brain Resource website as well as the direct-to-download dataset we used for our example workflow. The code used in the manuscript can be found at this GitHub repo https://github.com/viko18/JunaChimp (copy archived at https://archive.softwareheritage.org/swh:1:rev:411f0610269416d4ee04eaf9670a9dc84e829ea0/).

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
