## [Decision Letter]

**Acceptance summary:**

This paper introduces a chimpanzee brain reference template using structural brain scans, along with the code and processing pipeline necessary for implementing the process. Using this framework the authors also observe age-related changes in gray matter and hemispheric asymmetry in chimpanzee brains that broadly mirror trends seen in human studies. The work represents an important step forward in comparative neuroanatomy and a useful resource for the field.

**Decision letter after peer review:**

Thank you for submitting your article "Chimpanzee Brain Morphometry Utilizing Standardized MRI Preprocessing and Macroanatomical Annotations" for consideration by *eLife*. Your article has been reviewed by three peer reviewers, and the evaluation has been overseen by a Reviewing Editor and Timothy Behrens as the Senior Editor. The following individuals involved in review of your submission have agreed to reveal their identity: Leah A Krubitzer (Reviewer #1); Katherine L Bryant (Reviewer #2).

The reviewers have discussed the reviews with one another and the Reviewing Editor has drafted this decision to help you prepare a revised submission.

Summary:

Chimpanzees are our close relatives and their brains can help inform our understanding of human brain organization. Here a reference chimpanzee brain template is developed, and used to examine effects of aging and asymmetry. The atlas advances our understanding of the chimpanzee brain and is a valuable tool for comparative neuroanatomy.

Enthusiasm for the work was generally high. This is an important undertaking, and the transparent availability of the code was much appreciated.

Essential revisions:

1) If possible, it would be nice to see more detail regarding the comparative chimpanzee/human analyses (i.e., both symmetry and age-related decline). At minimum, some additional details in the text listing specific areas that are in common would be useful. If there is a way to link cross-species data in a quantified manner, that would be even better.

2) Understanding the limitations of the data, there was some concern about the use of sulcal boundaries and arbitrary subdivisions of the Davi130 parcellation. Are there other considerations that might be used in defining regions? Perhaps even explicitly addressing this issue would be useful. The parcellation is important not only for how we think about chimpanzee neuroanatomy, but as it directly effects the analyses and results. (This concern is helped somewhat by the voxel-based analyses.)

---

## [Author Response]

Essential revisions:1) If possible, it would be nice to see more detail regarding the comparative chimpanzee/human analyses (i.e., both symmetry and age-related decline). At minimum, some additional details in the text listing specific areas that are in common would be useful. If there is a way to link cross-species data in a quantified manner, that would be even better.

We agree that more detail regarding the specific common regions that present age-related decline and GM lateralization will be beneficial for the reader to better understand the inter-species similarities and differences. Therefore, we have elaborated upon our human comparison in the Discussion and explicitly stated the comparable human – chimpanzee regions for each of the references we provide (see paragraph three and paragraph nine of the Discussion for aging and asymmetry respectively).

To address the point of a cross-species quantitative comparison we computed the age-related effect on global GM (Figure 3D) in a matched human sample. We choose the IXI MRI dataset (https://brain-development.org/ixi-dataset/) for comparison as it contains both 1.5T and 3T images with a large age range. The human sample was matched to the QC passed chimpanzee sample using the ‘MatchIt’ R package (https://cran.r-project.org/package=MatchIt) by matching age, sex, and scanner field strength optimal the ‘optimal’ matching technique. As life span and the aging process is different between species the human sample was match to 1.5* chimpanzee age (Robson et al., 2008). The matched human sample distribution is illustrated in Figure 3C. The analysis shows that the matched human sample presents a more prominent aging effect with less variance as compared to the chimpanzees. Ideally, we would register the chimpanzee GM volume aging effects to the human template space to directly compare the spatial distribution of age effects, but this inter-species volume transformation is by no means trivial. We are currently working on this problem but are not yet able to present a working implementation.

Aging, Discussion- “…Additionally, there was no significant difference in the age-related decline between humans and chimpanzee (Figure 3), even though a larger negative correlation with less variance was found in the matched human sample as compared to the chimpanzees demonstrates a commonality in the healthy aging process that was thought to be specific to humans. […] Similar age-related total GM decline along with presentation in homologous brain areas suggests common underlying neurophysiological processes in humans and chimpanzees due to shared primate evolution.”

Asymmetry, Discussion – “A substantial amount of regions presenting significant inter-hemispheric differences of local morphology in chimpanzees has also been shown in humans (Good et al., 2001a; Plessen et al., 2014; Kong et al., 2018). […] This general organizational pattern of asymmetry was not apparent in the chimpanzee (Li et al., 2018).”

2) Understanding the limitations of the data, there was some concern about the use of sulcal boundaries and arbitrary subdivisions of the Davi130 parcellation. Are there other considerations that might be used in defining regions? Perhaps even explicitly addressing this issue would be useful. The parcellation is important not only for how we think about chimpanzee neuroanatomy, but as it directly effects the analyses and results. (This concern is helped somewhat by the voxel-based analyses.)

The concerns about the boundaries and subdivisions of regions in the Davi130 parcellation are very much warranted as these delineations do seldomly align with known microanatomical boundaries (Amunts and Zilles, 2015). As we are constrained by the properties of the MRI acquisition method, especially the limited resolution of the T1-weighted images, we kept our regions relatively large. In general, we utilized gross macroanatomical landmarks when present for region delineation and subdivision such as, gyri and sulci peaks, fundi, and bends. We have now further elaborated on the exact landmarks we used for region subdivision in the Davi130 parcellation section of the Materials and methods. Moreover, we argue that the Davi130 parcellation, as a reduced representation of the chimpanzee brain based on macroanatomical features within a standardized space facilitates future inter-species and chimpanzee region-wise brain morphology and association to behavior (see paragraph two of the Discussion).

Subdivisions, Materials and methods – “Considering the limitations of macroscopic features present in T1w we utilized distinct morphological representations to split large gyri, such as gyral/sulcal folds and continuation of sulci. If a distinguishable feature could not be determined, rough distance and regional size was employed as border defining criteria. […] Finally, splits within the OFC, SFG, and insula were based on equal size and/or distance. “

Davi130, Discussion – ”Additionally, we provide the manually segmented, macro-anatomical Davi130 whole-brain parcellation comprising 130 cortical, sub-cortical and cerebellar brain regions, which enables systematic extraction of volumes-of-interest from chimpanzee MRI data. […] Furthermore, our macroscopic parcellation mitigates the curse of dimensionality for multivariate machine learning methods to be applied to the relatively small samples like the NCBR.”